# Fabricating Ultra-Narrow Precision Slit Structures with Periodically Reducing Current Over-Growth Electroforming

**DOI:** 10.3390/mi15010076

**Published:** 2023-12-29

**Authors:** Xiaohong Yang, Xinmin Zhang, Pingmei Ming, Yuntao Li, Wei Wang, Yunyan Zhang, Zongbin Li, Lunxu Li, Youping Xiao, Xiaoyi Guo, Zheng Yang

**Affiliations:** 1School of Mechanical and Power Engineering, Henan Polytechnic University, Jiaozuo 454003, China; yangxiaohong@hpu.edu.cn (X.Y.); 15230636967@163.com (Y.L.); wangwei119230@163.com (W.W.); yun9404@163.com (Y.Z.); 18503843886@163.com (Z.L.); lilunxu1998@gmail.com (L.L.); 15215287591@163.com (Y.X.); tearsofsun@163.com (X.G.); yzheng1013@126.com (Z.Y.); 2School of Engineering and Technology, Henan University of Technology, Hebi 458030, China; 3School of Mechanical and Electronic Engineering, Hebi Polytechnic, Hebi 458030, China

**Keywords:** periodically reducing current, over-growth electroforming, ultra-narrow precision slit structure

## Abstract

An ultra-narrow precision slit with a width of less than ten micrometers is the key structure of some optical components, but the fabrication of these structures is still very difficult to accomplish. To fabricate these slits, this paper proposed a periodically reducing current over-growth electroforming process. In the periodically reducing current over-growth electroforming, the electric current applied to the electrodeposition process is periodically stepped down rather than being constant. Simulations and experimentation studies were carried out to verify the feasibility of the proposed process, and further optimization of process parameters was implemented experimentally to achieve the desired ultra-narrow precision slits. The current values were: $I_{1}=I_{initial},\;I_{2}=0.75I_{initial}\;at\;Q_{c}=0.5Q_{t},\;I_{3}=0.5I_{initial}\;at\;Q_{c}=0.75Q_{t},respectively.\;$It was shown that, compared with conventional constant current over-growth electroforming, the proposed process can significantly improve the surface quality and geometrical accuracy of the fabricated slits and can markedly enhance the achievement of the formed ultra-narrow slits. With the proposed process, slits with a width of down to 5 ± 0.1 μm and a surface roughness of less than 62.8 nm can be easily achieved. This can improve the determination sensitivity and linear range of the calibration curves of spectral imagers and food and chemical analysis instruments. Periodically reducing current over-growth electroforming is effective and advantageous in fabricating ultra-narrow precision slits.

## 1. Introduction

A micro-scale slit structure is used as the primary component with the function of light filtering, collimation, and diffraction [1,2,3,4], and the slit width directly affects the sensitivity and resolution in the field of spectrometry [5,6,7]. The slits for optical applications are often several micrometers in width and tens of millimeters in length. Generally, the narrower the width, the higher resolution and sensitivity the slits have. Correspondingly, the ultra-narrow slits with widths of less than 10 μm are required to have an extremely high geometrical accuracy including a very small dimensional tolerance and a very small surface roughness of slit walls. However, such ultra-narrow precision metal slits are significantly harder to fabricate.

To fabricate the ultra-narrow precision metal slits, several methods including conventional machining methods and nontraditional machining methods have been employed. They mainly involve mechanical micro-milling [8,9,10,11], laser beam machining [12,13,14,15,16,17,18], electro-discharge micromachining (Micro-EDM) [19,20], electron beam machining [21], electrochemical micromachining (Micro-ECM) [22], chemical milling [23,24], micro-electroforming (Micro-EF) [25,26,27], etc. For example, Xia et al. [28] utilized laser-induced oxidation-assisted micro-milling to fabricate high aspect ratio micro-slits on Ti6Al4V with a width of 0.5 mm and an aspect ratio of 5.4. Chow H M et al. [29] developed a modified rotating disk electrode (RDE) for the fabrication of micro-slit machining using EDM and obtained a 42 µm wide micro-slit on the Ti-6Al-4V workpiece. The above methods generate micro-slits by removing materials from the bulk workpiece based on the principles of mechanical shearing or thermal melting. These methods unavoidably leave debris on the slit wall and/or cause the machined thin-thickness slit to deform, so they are seldom used for ultra-narrow slits. Kim B H et al. [30] proposed an ultrashort ECM to fabricate slit microstructures but only achieved a tapered micro-slit whose upper width was 60 μm and lower width was 40 μm. Bing Zhu et al. [31] proposed an innovative micro-ECM process, i.e., wire-ECM, and successfully fabricated a 30-aspect-ratio 0.16 mm-wide slit. Although the micro-ECM processes based on an anodic dissolution mechanism do not cause debris or deformation defects, they are hard to generate high-accuracy ultra-narrow slits with due to the inherent stray corrosive effect. This indicates that an electrochemical process can provide a relatively desirable mechanism to generate micro-scale structures. Unlike ECM, which is a subtractive manufacturing process, micro-electroforming (micro-EF), which is also based on the electrochemical reaction process, uses the material additive process to create microstructures and micro-components atom by atom and layer upon layer. These unique machining mechanisms and forming processes enable micro-EF to extremely accurately fabricate or replicate the structures and articles. According to the mask used, micro-EF has two fabrication manners: maskless EF and mask EF. Among them, mask EF has a considerably higher formation accuracy, and so it is used more widely. Mask EF can be further divided into two categories: through-mask electroforming (TM-EF) and over-growth electroforming (over-growth EF). TM-EF forms microstructures and micro-components by filling the reduced atoms into the predefined through-mask cavities (molds), which are geometrically accurate, finally achieving the replicates of the through-mask molds. However, this kind of EF is hard to create ultra-narrow slits with because the corresponding ultrathin convex photoresist molds are hard to make. Unlike TM-EF, over-growth EF is mostly used to generate tapered hollow microfeatures such as micro-sized conical orifices and tapered slits due to its different formation manner. The total procedure of over-growth EF involves two phases. In the first phase, the deposited materials are gradually filled into the photoresist mold cavities, as in TM-EF, until the cavities are completely filled up, and then the second phase starts. In the second phase, the deposited materials not only continue to grow along the direction of the thickness but also expand freely along the horizontal direction on the top surface of the photoresist masks, thus gradually forming the narrowing hollow structures by the expanding deposited materials and finally giving a tapered hollow structure—lower big and upper small. With this nontraditional EF, some micro-scale precision hollow structured components have been created. Yunyan Zhang et al. [32] used this technique to fabricate a 3 μm-diameter micro-hole plate for a precision medical atomizer. The fabricated micro-holes have a favorable dimensional tolerance, which is within ±0.1 μm. However, ultra-narrow slits with a width of less than ten micrometers are significantly difficult to fabricate when the standard over-growth EF is used. This is probably because the mass transportation conditions become poorer and poorer with the narrowing of the forming slits or holes, generating more defects. To overcome these drawbacks, a modified over-growth EF process was proposed. In the modified over-growth EF, a gradually reducing interelectrode (between the anode and cathode) electric current rather than a constant electric current was used during the electrodeposition, so it was called periodically reducing current over-growth electroforming (PRC over-growth EF). With PRC over-growth EF, ultra-narrow precision slits with widths of several micrometers are expected to be fabricated. In the following sections, PRC over-growth EF is modeled via a numerical simulation based on the tertiary current distribution and validated experimentally and further attempts are made to fabricate the components with ultra-narrow slits.

## 2. Principle and Simulation Analysis of PRC Over-Growth EF

### 2.1. Principle of PRC Over-Growth EF

The schematic diagram of PRC over-growth EF for fabricating slit structures is shown in Figure 1. In PRC over-growth EF, the interelectrode electric current, *I*, applied to the electrodeposition process is periodically reduced according to some criterion rather than kept constant. For example, *I* is reduced two times according to the increase in Coulombic quantity consumed, i.e., $I_{1}=I_{initial},\;I_{2}=0.75I_{initial}\;at\;Q_{c}=0.5Q_{t},\;I_{3}=0.5I_{initial}\;at\;Q_{c}=0.75Q_{t}.\;$Hereto, $I_{initial}$ is the initial value of the interelectrode electric current applied to the process, $Q_{t}$ is the theoretical value of the total Coulomb volume of the whole electroforming process. The main reason for periodically reducing current during electrodeposition is to reduce the possibility of mass transportation limitation because the depositing space where the electrochemical reactions are taking place increasingly narrows with deposition time.

### 2.2. Simulation Analysis

#### 2.2.1. Simulations

In order to understand the variation in electrolyte flow rate, cathode current density, and cation concentration with electrodeposition time while forming ultra-narrow slits with PRC over-growth EF, a two-dimensional simulation model based on the tertiary current distribution was developed, as shown in Figure 2. The upper boundary denotes the anode, Γ_1_, and the lower boundary denotes the cathode, Γ_3_. The cathode is covered with well-defined photoresist mask patterns, which are denoted by Γ_4_. The left boundary of the model is defined as the electrolyte inlet and the right boundary is defined as the electrolyte outlet, denoted by Γ_2_ and Γ_5_, respectively.

Since the electroforming process involves multiphysics fields including an electrochemical reaction field, an electric current field, a mass transfer rate field, an ion concentration field, etc. [33], multiple field theories are theoretically needed to simulate such a process. However, this is extremely difficult. Usually, a simplified electric field and flow field coupled multiphysics field are used to facilitate the simulation. For this, some assumptions were made as follows.

(1)Boundary effects are neglected and the potentials at each electrode surface are equal.(2)The electrolyte is isotropic and its conductivity is constant over the confined area.(3)Concentration polarization is not considered, and only the mass transport of nickel cations associated with electrodeposition is considered in the simulations. The anode boundary is set as an invariant boundary, while the cathode boundary is a geometric deformation boundary.

In the numerical model, the mass transportation of nickel flux in the electrolyte is expressed as follows:(1)$$N_{\mathrm{Ni}}=-D_{\mathrm{Ni}}\nabla c_{\mathrm{Ni}}-z_{\mathrm{Ni}}u_{\mathrm{Ni}}Fc_{\mathrm{Ni}}\nabla \phi_{l}$$
(2)$$\frac{\partial c_{Ni}}{\partial t}+\nabla \cdot N_{Ni}=0$$
where *N_Ni_* denotes the transport vector (mol/(m^2^·s)), $c_{Ni}$ is the nickel concentration in the electrolyte (mol/m^3^), z*_Ni_* is the charge of the nickel ion, *u_Ni_* is the mobility of the nickel ion (m^2^/(s·J·mol)), *F* is Faraday’s constant (C/mol), and ${\;\phi }_{l}$ is the electrolyte potential (V).

The current density is obtained by solving the above equations with the electrolyte conductivity *σ_l_* and the electrolyte potential${\;\phi }_{l}$. At the cathode–electrolyte interface, the local current density *i_loc_* is given by the Butler–Volmer equation.
(3)$$i_{l}=-\sigma_{l}\nabla {\;\phi }_{l}$$
(4)$$i_{loc}=i_{0}\left(exp\left(\frac{\alpha_{a}F\eta }{RT}\right)-exp\left(\frac{\alpha_{c}F\eta }{RT}\right)\right)$$
where *α_a_* is the anodic transfer coefficient, *α_c_* is the cathode transfer coefficient, *i*_0_ is the exchange current density, *η* is the overpotential, *R* is the gas constant, and *T* is the Kelvin temperature.

The deformation caused by electrodeposition was controlled by the local current density and nickel icon transportation, and the equation was established using Faraday’s law:(5)$$N_{Ni}=-\frac{i_{loc}}{2F}$$
(6)$$V_{n}=-\frac{i_{loc}}{F}\frac{M}{z\rho_{M}}$$
where *V_n_* is the deformation rate, *M* is the mean molar mass, *ρ* is the nickel density, and *n* is the number of participating electrons.

The same model was employed to simulate the slit formation under different current modes, i.e., the constant current mode and the periodically reducing current mode, in which the total charge on the anode was kept constant while different current densities were applied at the anode boundary. In the constant current mode, the current was maintained at the level of *I* = 1 A. Specifically, the periodically reducing current mode was divided into three phases, as shown in Figure 1c. The definition and boundary conditions of the simulation model are listed in Table 1.

#### 2.2.2. Results and Discussion

(1)Flow field distribution

The velocity variation during the over-growth EF simulation for the slit formation process is shown in Figure 3. It was found that the simulated geometric deformation was mainly concentrated above the photolithography mask, as the deposition height increased from 21 μm to 48 μm while the slit width decreased from 50 μm to 5 μm, as shown in Figure 3b. Correspondingly, the velocity distribution near the slit shifted with the raised contour profiles. At the same time, the electrolyte filled into the slit was slowed down to less than 0.005 m/s, which can be considered stagnant. The formation of ultra-narrow slits resulted in an extremely low electrolyte flow within the narrowing slit. The space for metallic ion transportation became increasingly restricted. Despite attempts to increase the electrolyte flow rate over the cathode, the convective mass transfer efficiency within the extremely narrow slits was not significantly improved.

(2)Cathodic current density distribution

The cathodic current density distribution under the modes of the constant current and the PRC over-growth EF current is shown with contours ranging from 0 to 120 A/m^2^ in Figure 4. The same slit-forming shapes were found as the slit shrunk. However, the distribution of current density changed more significantly according to the slit’s structural formation as the slit width changed from 50 μm to 5 μm. The formation caused the concentrated cathodic current density on the tapered slit wall. Specifically, in the constant current mode, the maximum current density at the surface of the slit structure reached 162 A/m^2^ when the slit width was 5 μm, and the current density at the bottom of the slit was almost negligible, while the flat sides had a constant current density of 100 A/m^2^. On the contrary, the maximum current density at the slit decreased to 65 A/m^2^ in the PRC mode. Compared with the constant current mode, the periodically reducing current mode can significantly improve the current density concentration on the cathode surface, which is favorable for obtaining high-quality slit structures.

(3)Ion concentration distribution

The ion concentration distributions of the over-grown EF with constant current and periodically reduced current modes are shown in Figure 5. When the slit width was reduced from 50 μm to 5 μm using the constant current mode, the ion concentration at the slit decreased significantly. At a slit width of 5 μm, the lowest ion concentration was 224 mol/m^3^, which was less than half of the electrolyte bulk concentration. The lowest ion concentration was 440 mol/m^3^ at a slit width of 5 μm when using the PRC current mode. The simulation result has shown that the PRC current contributed to the improved ion concentration at the bottom, which was beneficial for the geometrical accuracy of the ultra-narrow slit structure under poor mass transportation conditions.

The simulation results show that the mass transfer behavior of metal cations from the electrolyte to the cathode surface during the slit formation using the over-growth electroforming electrodeposition technique deteriorates as the slit width decreases, as shown in Figure 6. The process is categorized into three phases including a phase of good mass transfer (Figure 6a), near the phase where mass transfer is limited (Figure 6b), and a mass transfer limitation phase. In the good mass transfer phase, the slit size is wide and the migration of cations is not restricted. However, at the stage where mass transfer is limited, the decrease in slit width hinders cation migration into the slit structure, resulting in a lower cation concentration in the region with a narrower slit width. In addition, the cathode undergoes a hydrogen reaction to produce bubbles, which can affect the electrodeposition quality. A periodically reducing current model using over-growth EF can enhance mass transfer in the fabrication of ultra-narrow slit structures. This is because the input current is periodically lowered, resulting in a subsequent decrease in cationic demand at the cathode, which improves the ion-deficient state of the slit region under mass-transfer-limited conditions.

## 3. Experimental Study

### 3.1. Materials and Methods

A 304 stainless steel plate with a thickness of 1 mm was selected as the substrate. The cathode substrate was covered by the photolithography mask array with a thickness of 3 μm, a width of 80 μm, and a length of 3 mm. The experiments used the nickel sulfamate system, electrolyte ratio, and experimental process parameters shown in Table 2. The surface morphology detection equipment used is FEI-F50 Nova450 field emission scanning electron microscope (SEM, Beijing, China), the surface roughness detection equipment is the confocal laser scanning microscope (CLSM) type IX83-FV3000 (Olympus, Tokyo, Japan), and the geometrical accuracy detection was performed using the AM600CNC imaging instrument (Dongguan, China).

In order to compare and analyze the quality of the slit structure, two modes were used: the constant current mode and the periodically reducing current mode over-growth EF. Combining the electrodeposition efficiency and molding quality, a constant current model with an interpolar current of 1 A and a periodic reduced current mode with interpolar currents of 1 A, 0.75 A, and 0.5 A were used. According to the theoretical total Coulomb amount *Q_t_* required to fabricate the slit structure and the molding characteristics, the whole fabrication process was divided into three stages according to the Coulomb variation and the molding characteristics, and the interpolar current was constant at each stage. The detailed stages of changing the interpolar current are when the Coulomb quantity increases from 0 to 0.5 *Q_t_*, the interpolar current is 1 A; when the Coulomb quantity increases from 0.5 *Q_t_* to 0.75 *Q_t_*, the interpolar current is 0.75 A; and when the Coulomb quantity increases from 0.75 *Q_t_* to *Q_t_*, the interpolar current is 0.5 A.

### 3.2. Results and Analysis

#### 3.2.1. Morphology and Surface Quality

The ultra-narrow slit structure prepared by PRC over-growth EF is shown in Figure 7. The component array layout and single slit structure are shown in Figure 7a,b, respectively. The camera photograph is shown in Figure 7. The dimensions shown in the figures are the feature dimensions of the slit component.

The SEM image of the slit is shown in Figure 8. In the over-growth EF process, as the slit width decreases, the mass transfer limitation at the slit becomes more and more serious. At this time, if the current input value at the stage of good mass transfer continues to be used, i.e., the magnitude of the current input value remains unchanged, the metal cations generated at the anode will not be able to reach all the way to the cathode due to the mass transfer limitation, and the phenomenon of metal cation excess will occur in the solution. When using PRC over-growth EF, as the slit width is gradually smaller, the current input value is gradually reduced according to the mass transfer limitation. At this time, the number of metal cations supplied by the anode in the unit time is reduced, which is exactly matched with the number of cations that can reach the cathode and undergo the electrodeposition reaction, and there will be no excess cations in the solution so that the phenomenon of nodules due to the deposition of metal cations in the rest of the slit is avoided. The phenomenon of nodulation due to the deposition of metal cations in other parts of the slit will be avoided.

The surface roughness test results are shown in Figure 9. It was found that the surface roughness value of the PRC over-growth EF slit structure (62.8 nm) was lower than those formed by constant current over-growth EF (72.6 nm).

#### 3.2.2. Geometrical Accuracy

The AM600CNC imaging instrument (Dongguan, China) is used to observe the slit structure. The slit parts are placed horizontally on the imaging instrument table, the intensity of the light source above and below the lens of the imaging instrument is adjusted, and the part that transmits light, which is the slit part, is in a bright state, and the part that does not transmits light is the outer edge of the slit, and the slit part is the outer edge of the slit. As shown in Figure 10a, the two edge lines of the slit made using the constant current mode are uneven and jagged, showing a sawtooth shape, and there are more nodules. The two edge lines of the slit utilizing PRC over-growth EF are straight, with a complete shape and uniform growth, as shown in Figure 10b. The results show that the poor quality of the slit molding is due to the uneven mass transfer of the electrolyte during electroforming when the constant current mode is used, and as the slit width becomes narrower, the mass transfer becomes worse, resulting in uneven growth of the slit edges and the formation of jagged edges. As shown in Figure 10a, the two edge lines of the incision using PRC over-growth EF were straight, with a complete shape and uniform growth, and the accuracy of the slit width ranged from ±0.1 μm. The two edge lines of the slit fabricated using the constant current mode were uneven, with more nodules, and the slit widths were uneven with a wide range of variations, as shown in Figure 10b. The main reason for this is the same as the effect of the current mode on the surface morphology.

In order to characterize the straightness and parallelism of the slit, a reference coordinate system was established with the length of the slit parallel to the *x*-axis, as shown in Figure 11a. The distances from both edges of the slit to the *x*-axis were measured: y_s_ for the distance from the upper edge to the *x*-axis, and y_x_ for the distance from the lower edge to the *x*-axis. Fifty points on both edges were measured and plotted as scatter plots, as shown in Figure 11b,c. As shown in Figure 11b, the points with a distance of 5 μm between the slits were neatly arranged in a straight line, which indicates that the parallelism of the two slits is good, while in Figure 11c, it is shown that the points at the wall of the slits deviate from the straight line, which indicates that the parallelism of the slits is poor. By observing these scatter plots, it is clear that both edges of the slit exhibit high straightness while maintaining high parallelism under the mode of PRC over-growth EF.

In the above, two electrochemical deposition modes (constant current mode and periodically decreasing current mode) were used to prepare slit structures with a slit width of 5 μm, and the surface quality, surface morphology, and geometrical accuracy of the slit structures were characterized. Compared with the constant-current mode, slits with a small range of slit size variation, uniform size, and which are flat can be obtained by using the periodic decreasing current mode. The dimensions varied between 0.0049 and 0.0051 mm and the slits were free from defects such as nodules and sawtooth shapes. It shows that the surface roughness value at the slit is even smaller and can reach 62.8 nm with good surface quality. For the above phenomenon, the paper reveals the forming principle of periodic decreasing current mode over electroforming from the principle of electrodeposition slits, i.e., as the width of the slit decreases, the amount of metal cations provided by the anode to the cathode is limited, and a state of cation excess occurs. During the deposition process, if the state of excess metal cations can be avoided by matching the amount of metal ions supplied by the anode to the amount of metal ions deposited, there will be no excess metal ions and the nodule phenomenon can be avoided. Finally, it was found that PRC over-growth EF significantly improved the surface topography, surface quality, and geometrical accuracy of the slit structure.

## 4. Conclusions

To fabricate ultraprecision micro-slit structures using the electroforming process, specially ultra-narrow slits, in order to improve the resolution and sensitivity in filtering, collimation, and diffraction, a modified PRC over-growth EF technique was proposed. Numerical simulations and experimental studies were carried out to verify its feasibility and practicability. Some conclusions are made as follows.

(1)PRC over-growth EF is significantly effective and advantageous for fabricating ultranarrow precision slits, since it can maintain a constantly favorable condition for the total electrodeposition process.(2)With PRC over-growth EF, slits with a width of down to 5 ± 0.1 μm and a surface roughness of less than 62.8 nm can be easily achieved.(3)With PRC over-growth EF, the reduction in the current is carried out preferentially in terms of the consumed Coulomb quantity.

## Figures and Tables

**Figure 1 micromachines-15-00076-f001:**
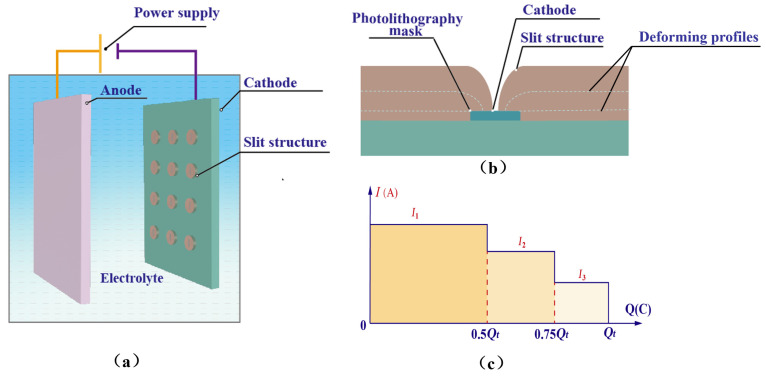
Schematic diagram of the periodically reducing current over-growth electroforming for fabricating slit structures. (**a**) Technical approach. (**b**) Process of over-growth EF. (**c**) Periodically reducing current.

**Figure 2 micromachines-15-00076-f002:**
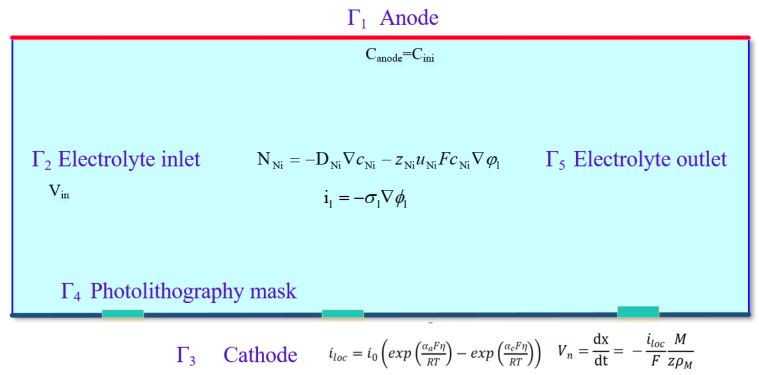
Two-dimensional model established to simulate PRC over-growth EF.

**Figure 3 micromachines-15-00076-f003:**
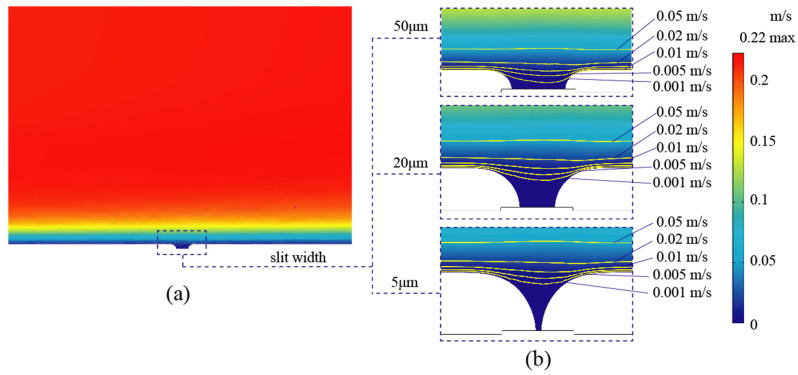
Velocity variation during the over-growth EF simulation. (**a**) The velocity distribution between the anode and cathode. (**b**) Local flow distribution with velocity contours as the slit width decreased from 50 μm to 5 μm.

**Figure 4 micromachines-15-00076-f004:**
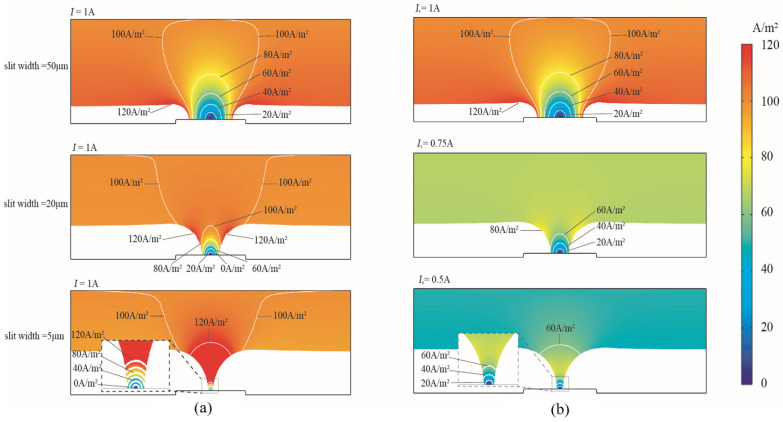
Cathodic current density distribution of over-growth EF. (**a**) Constant current mode. (**b**) Periodically reducing current mode.

**Figure 5 micromachines-15-00076-f005:**
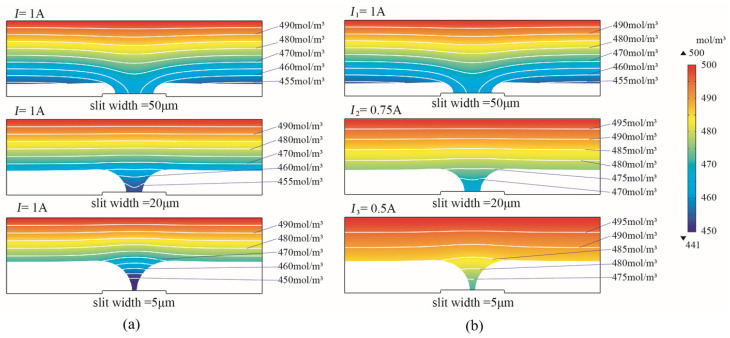
Ion concentration distribution of over-growth EF. (**a**) Constant current mode. (**b**) Periodically reducing current mode.

**Figure 6 micromachines-15-00076-f006:**
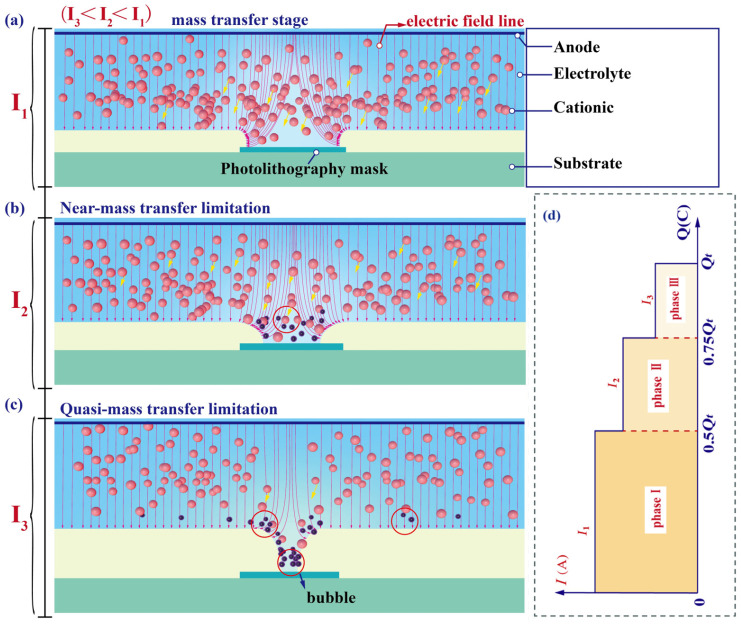
Schematic diagram of mass transfer during over-growth EF. (**a**) Phase of good mass transfer. (**b**) Near the phase where mass transfer is limited. (**c**) Phase in which mass transfer is limited. (**d**) Phase of periodically reducing currents corresponding to electrodeposition charge.

**Figure 7 micromachines-15-00076-f007:**
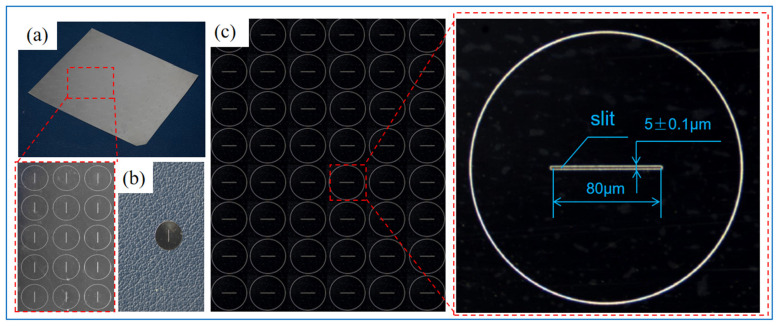
Total view of the ultra-narrow slit structure prepared by PRC over-growth EF. (**a**) The total view of the component with slit array. (**b**) Partial view of the slit array. (**c**) The dimensions and accuracy of the formed slit.

**Figure 8 micromachines-15-00076-f008:**
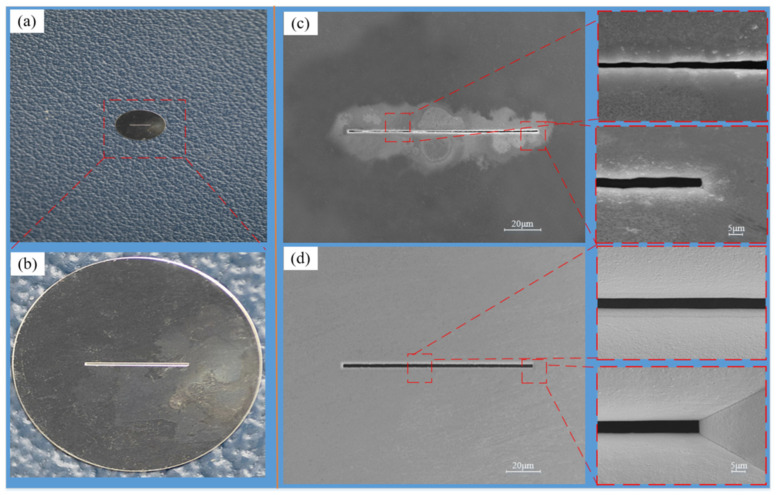
The component with the slit structure and the SEM image of the slit structure. (**a**) Photographs of individual slit parts. (**b**) Enlarged view of a single slit part. (**c**) Constant current mode. (**d**) Periodically reducing current mode.

**Figure 9 micromachines-15-00076-f009:**
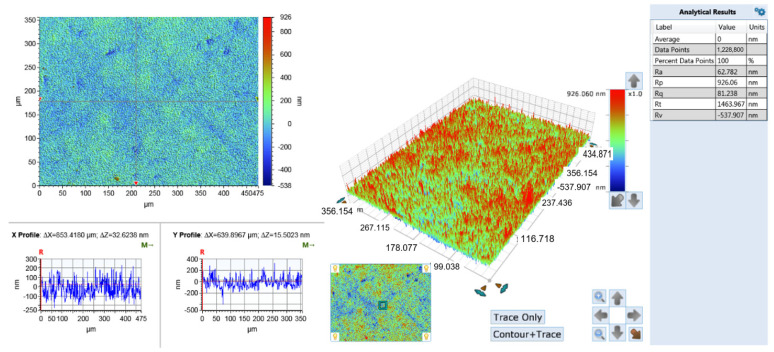
Characterization of slit surface roughness in PRC over-growth EF mode.

**Figure 10 micromachines-15-00076-f010:**
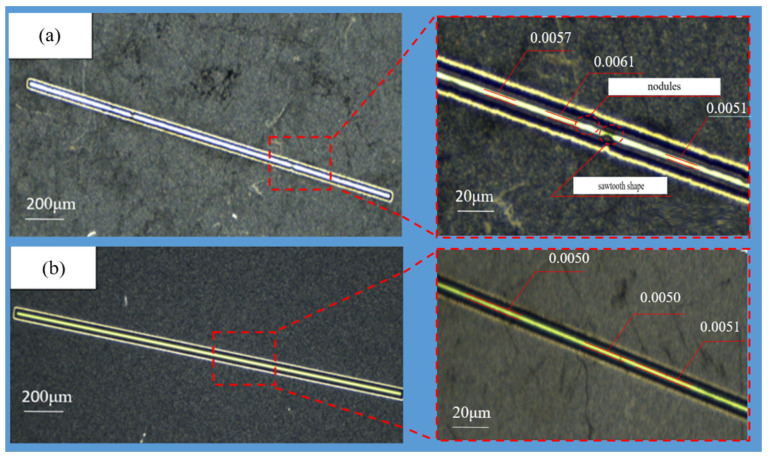
The enlarged photo view of a single slit’s structure. (**a**) Constant current mode. (**b**) Periodically reducing current mode.

**Figure 11 micromachines-15-00076-f011:**
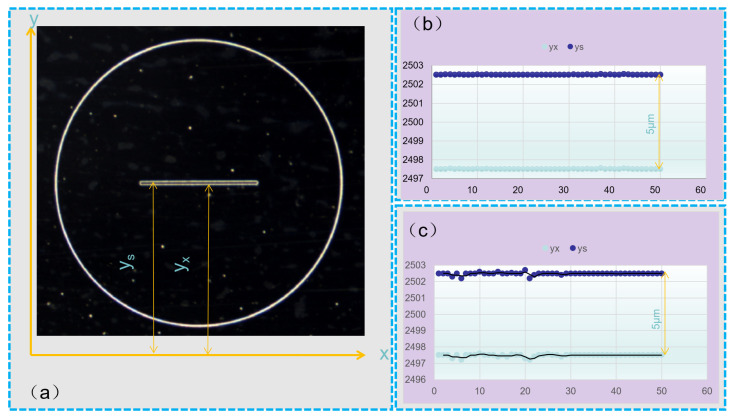
Characterization of the straightness and parallelism of slits. (**a**) Coordinate systems for evaluating edge accuracy. (**b**) Periodically reducing current mode. (**c**) Constant current mode.

**Table 1 micromachines-15-00076-t001:** Definition and boundary conditions of the simulation model.

Notation	Description	Property
Γ_1_	Anode	$\mathrm{Ion}\;\mathrm{concentration}\;C{\mathrm{anode}}_{\;}=500\;\mathrm{mol}/m^{3}$ Constant current mode: I=1 A Periodically reducing current mode: $Q_{c}=($$0–0.5)Q_{t}$$,\;I_{1}=1\;A$ $Q_{c}=($$0.5–0.75)Q_{t}$$,\;I_{2}=0.75\;A$ $Q_{c}$= (0.75–1)$Q_{t}$, *I*_3_ = 0.5 A
Γ_2_	Electrolyte inlet	*V*_in_ = 0.2 m/s
Γ_3_	Cathode	$\mathrm{Set}\;\mathrm{as}\;\mathrm{the}\;\mathrm{equipotential}\;\mathrm{boundary}\;\mathrm{with}\;0\;V\;\mathrm{and}\;\mathrm{the}\;\mathrm{geometrical}\;\mathrm{deformation}\;\mathrm{rate}\;\mathrm{depicted}\;\mathrm{as}\;V_{n\;}$
Γ_4_	Photolithography mask	No flux
Γ_5_	Electrolyte outlet	-

**Table 2 micromachines-15-00076-t002:** Electrolyte composition and experimental process parameters.

Items (Unit)	Value
Nickel sulfamate (Ni(SO_3_NH_2_)_2_·4H_2_O)/(g/L)	500
Nickel chloride (NiCl_2_·6H_2_O)/(g/L)	6–10
boric acid (H_3_BO_3_)/(g/L)	30–40
Temperature (°C)	50–55
pH	3.8–4

## Data Availability

Data available on request from the authors. The data that support the findings of this study are available from the corresponding author, [P.M.], upon reasonable request.

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
