# Peer review of "Fabricating Ultra-Narrow Precision Slit Structures with Periodically Reducing Current Over-Growth Electroforming"

_micromachines, 2023, doi:10.3390/mi15010076_

Round 1

Reviewer 1 Report

Comments and Suggestions for Authors

1.       It would be beneficial to include specific details about the simulations and experimental studies conducted to verify the proposed process. Describing the key parameters investigated and their impact on the fabrication process could enhance the clarity of the abstract.

Author Response

 I sincerely thank you for taking your time to review this review, and thank you very much for your comments. We have carefully considered your opinions in the revised version. We have revised them one by one according to your suggestions and now reply as annex.

I would like to resubmit this manuscript to Micromachines””, and hope it is acceptable for ublication in the journal. If there are any problems or questions about our paper, please do not hesitate to let us know.

Thank you very much for your attention to our paper.

Sincerely yours,

Dr. Pingmei Ming and Xiaohong Yang

Reviewer 2 Report

Comments and Suggestions for Authors

Summary of review No. Micromachines-2772415

Fabricating Ultra-narrow Precision Slit Structures with Periodically Reducing Current Over-growth Electroforming

No. Micromachines2772415

This paper aims to explore the benefits of a modified periodically reducing current (PRC) over-growth electroforming technique for fabrication in ultraprecision microslit structure. In this article the numerical simulations and experimental studies were carried out to verify its feasibility and practicability. Moreover, the optimization of process parameters was also determined experimentally to achieve the desired ultra-narrow precision slits.

The substantial contents, organization, and the description of the manuscript are appropriate and satisfactory. The proposed process can significantly improve the surface quality and geometrical accuracy of the fabricated slits, and can markedly enhance the achievement of the forming ultra-narrow slits. On the whole, I find this paper can be considered for publication.

The following is the list of detailed comments with respect to the paper:

l The style of reference format listed at the end should be used consistently such as Nos.28, 31, 34, 35. Moreover, some of the reference cited in the main text should be revised carefully such as No.26 did not cite in the text, Nos. 23-27 are different to the references listed at the end, and the surname used only that cited in the text.

l Figures 4 and 5 should be indicated the classification (a) or (b).

l The English writing should be checked carefully especially the space within sentence.

l In 3.3.2 Geometrical accuracy, the descriptions with the results shown in Figure 10 are conflicted. and the Caption of Figure 10 should be checked carefully.

Comments on the Quality of English Language

-

Author Response

  I sincerely thank you for taking your time to review this article, and thank you very much for your comments. We have carefully considered your opinions in the revised version. We have revised them one by one according to your suggestions and now reply as annex.
